# Cardiac Sarcoidosis Is Uncommon in Patients with Isolated Sarcoid Uveitis: Outcome of 294 Cases

**DOI:** 10.3390/jcm10102146

**Published:** 2021-05-15

**Authors:** Mael Richard, Yvan Jamilloux, Pierre-Yves Courand, Laurent Perard, Cécile-Audrey Durel, Arnaud Hot, Carole Burillon, Isabelle Durieu, Mathieu Gerfaud-Valentin, Laurent Kodjikian, Pascal Seve

**Affiliations:** 1Department of Internal Medicine, Hôpital de la Croix-Rousse, Hospices Civils de Lyon, Université Claude Bernard Lyon 1, 69004 Lyon, France; mael.richard@chu-lyon.fr (M.R.); yvan.jamilloux@chu-lyon.fr (Y.J.); mathieu.gerfaud-valentin@chu-lyon.fr (M.G.-V.); 2Department of Cardiology, Hôpital de la Croix-Rousse et Hôpital Lyon Sud, Hospices Civils de Lyon, Creatis, Université Claude Bernard Lyon 1, 69004 Lyon, France; pierre-yves.courand@chu-lyon.fr; 3Department of Internal Medicine, Hôpital Saint-Joseph Saint-Luc, 69007 Lyon, France; lperard@ch-stjoseph-stluc-lyon.fr; 4Department of Internal Medicine, Hôpital Edouard Herriot, Hospices Civils de Lyon, 69003 Lyon, France; cecile-audrey.durel@chu-lyon.fr; 5Department of Internal Medicine, Hôpital Edouard Herriot, Hospices Civils de Lyon, Université Claude Bernard Lyon 1, 69003 Lyon, France; arnaud.hot@chu-lyon.fr; 6Department of Ophthalmology, Hôpital Edouard Herriot, Hospices Civils de Lyon, Université Claude Bernard Lyon 1, 69003 Lyon, France; carole.burillon@chu-lyon.fr; 7Department of Internal and Vascular Medicine, Hôpital Lyon Sud, Hospices Civils de Lyon, Université Claude Bernard Lyon 1, 69003 Lyon, France; isabelle.durieu@chu-lyon.fr; 8Department of Ophthalmology, Hôpital de la Croix-Rousse, Hospices Civils de Lyon, Université Claude Bernard Lyon 1, 69004 Lyon, France; laurent.kodjikian@chu-lyon.fr; 9Laboratoire UMR-CNRS 5510 Matéis, Université Lyon 1, 69100 Villeurbanne, France; 10Hospices Civils de Lyon, Pôle IMER, F-69003 Lyon, France; 11University Claude Bernard-Lyon 1, HESPER EA 7425, F-69008 Lyon, France

**Keywords:** sarcoidosis, uveitis, cardiac sarcoidosis

## Abstract

Recently, concerns have been raised about an increased risk of cardiac sarcoidosis in patients with sarcoid uveitis. While cardiac sarcoidosis has a high mortality burden, there is still a lack of precise data on this association. The objective of this study is to describe the frequency and type of cardiac complications associated with sarcoidosis of a large cohort of patients with sarcoid uveitis. We analyzed the cardiac outcomes of a monocentric retrospective cohort of consecutive adults with a diagnosis of sarcoid uveitis between January 2004 and March 2020 in a tertiary French university hospital. A total of 294 patients with a final diagnosis of sarcoid uveitis were included. At final follow-up, seven (2.4%) patients of the cohort had cardiac sarcoidosis. Cardiac sarcoidosis was more frequent among patients with previously reported systemic sarcoidosis (*p* = 0.008). The prevalence of cardiac sarcoidosis among patients with sarcoid uveitis is low, but patients with previously diagnosed sarcoidosis or those who develop systemic sarcoidosis during follow-up appear to be at increased risk.

## 1. Introduction

Uveitis is a frequent manifestation of sarcoidosis (22 to 63% of cases) [1,2,3] and is often a revelatory symptom of it [1,4]. Ocular involvement is most often isolated, but it can likewise be part of a multisystemic context [5]. The occurrence of symptomatic extraocular involvement in patients with initially isolated sarcoid uveitis has been reported in 7.7 to 17% of cases [5,6]. Cardiac involvement affects about 3 to 39% of the patients with systemic sarcoidosis [7,8,9,10,11,12,13]. However, an autopsied series of patients with systemic sarcoidosis revealed cardiac granulomas in greater proportions (in up to 46.9% of the cases) [14,15].

There is scant evidence of secondary cardiac involvement in the context of sarcoid uveitis. Recently, Han et al. reported on a series of 249 patients referred for etiological diagnosis of uveitis [10]. Of the 19 patients considered to have sarcoid uveitis, four (21%) presented with ventricular tachycardia, requiring an implantable cardioverter defibrillator (ICD). These results tend to support extensive screening due to this high prevalence. In addition, Schupp et al. [8] reported frequent ocular and cardiac involvement associated with dermatological and neurological involvement. It is of great importance to unveil potential cardiac sarcoidosis because, even though the disease may be paucisymptomatic, it is linked to significant morbidity and mortality [14,16,17,18]. Additional and precise data sets on concomitant or secondary cardiac involvement in patients with sarcoid uveitis are greatly needed. Because of this, the aim of this study was to describe the frequency and the type of cardiac complications in patients afflicted by sarcoid uveitis.

## 2. Materials and Methods

### 2.1. Study Population

Consecutive adult patients with definite or presumed sarcoid uveitis referred to the Departments of Ophthalmology and Internal Medicine of the Lyon University Hospital between January 2004 and March 2020 were retrospectively included. The Lyon University Hospital is a tertiary referral center for rare diseases, and most of the patients first referred to the Department of Ophthalmology for diagnosis of uveitis are then addressed to the Department of Internal Medicine for the etiological diagnosis work-up. In general, this population can be considered to be representative of urban populations in Western European areas.

The diagnosis of uveitis was assessed by ophthalmologists after a comprehensive eye examination, including biomicroscopic verification, assessment of the best corrected visual acuity, intraocular pressure, and, eventually, angiography or optical coherence tomography. The classification of uveitis was based on the Standardization of Uveitis Nomenclature [19]. Clinical signs suggesting ocular sarcoidosis were assessed by ophthalmologists. All patients were evaluated by an internist experienced in uveitis management and included at least a complete clinical examination, chest CT, or X-ray and basic laboratory tests.

### 2.2. Definitions

Biopsy-proven sarcoidosis was defined according to the WASOG/ATS/ERS criteria [20]. Presumed sarcoidosis was defined by the presence of two or more of the following: chest imaging showing abnormalities consistent with sarcoidosis; elevated angiotensin converting enzyme (ACE); lymphocytic alveolitis (presence of >15% lymphocytes and CD4/CD8 ratio > 3.5 in a bronchoalveolar lavage (BAL) fluid analysis) [21]. All patients with other granuloma-forming diseases, such as tuberculosis (including patients with biopsy-proven necrotizing granulomas and positive QuantiFERON-TB tests), were excluded, as were those with positive treponemal serology.

Pulmonary involvement was defined by respiratory symptoms and chest imaging (X-ray or CT), which were deemed compatible [20,22]. Bilateral hilar lymphadenopathy alone was not considered as pulmonary involvement but was used as a diagnostic criterion. Histological proof was required for diagnosing cutaneous sarcoidosis.

Cardiac symptoms were defined as: dyspnea with no pulmonary involvement, palpitations, syncope, and signs of heart failure (such as lower limb edema or lung crackles) [23]. An electrocardiogram (EKG) was defined as abnormal if any of the following were present: ventricular hypertrophy; left or right bundle branch block; presence of unexplained repolarization abnormality; any atrioventricular block (AVB); ventricular tachycardia. Cardiac magnetic resonance imaging (cMRI) was considered suggestive of cardiac sarcoidosis in case of patchy late gadolinium enhancement, and 18-FDG positron emission tomography (18-FDG PET) was considered suggestive of cardiac sarcoidosis in case of patchy pattern of uptake. Cardiac sarcoidosis was diagnosed when the patient fulfilled either the Japanese Circulation Society 2017 criteria [24] or the Heart and Rhythm Society 2014 criteria [11].

Patients were screened into three groups according to the revelatory symptoms and clinical presentation: (i) Group 1: clinically isolated ocular sarcoidosis (patients with uveitis as a presenting sign of sarcoidosis free of further involvement for at least 1 year of follow-up); (ii) Group 2: ocular sarcoidosis associated with systemic involvement (patients with uveitis as the presenting sign but afflicted with extraocular involvement at the time of diagnosis or within the first year of follow-up); and (iii) Group 3: systemic sarcoidosis (patients diagnosed with systemic sarcoidosis before the onset of uveitis). The secondary onset of cardiac and extraocular involvement was defined as taking place after 1 year of follow-up.

### 2.3. Data Collection

Data were extracted from the medical records of the patients. Two investigators (MR and PS) collected and analyzed clinical, ophthalmological, imaging and laboratory data using a standardized form, with particular attention paid to cardiac parameters, examinations, and outcomes: cardiac and respiratory symptoms, EKG, transthoracic echocardiography (TTE), cMRI, 18-FDG PET, local and systemic treatments, and ICD requirement. All available EKG results were reviewed.

### 2.4. Statistical Analyses

Categorical data are expressed as numbers and percentages and were compared using Fisher’s exact test. Continuous variables are described as the mean (± standard deviations) or median and interquartile ranges. Continuous variables were compared using the non-parametric Kruskal–Wallis test. All analyses were performed using R-software V.4.0.2 (R Foundation for Statistical Computing, Vienna, Austria). Statistical significance was set at a level of *p* = 0.05.

### 2.5. Ethical Considerations

This study received approval from the local ethics committee in March 2019 (No. 19-31) and was registered on clinicaltrials.gov (NCT03863782). According to French law, written consent for this study was not required due to its retrospective nature.

## 3. Results

### 3.1. Baseline Characteristics of the Population

During the study period, 294 patients with an established diagnosis of sarcoid uveitis were referred to our teaching hospital. The characteristics of the study population and the final diagnoses are shown in Table 1. The mean age at uveitis onset was 51.1 (±18.3) years. A total of 195 (66%) patients were female and 185 (63%) were of Caucasian ancestry. In total, 218 (74%) patients had biopsy-proven sarcoid uveitis and 76 (26%) patients had presumed sarcoid uveitis. Following the SUN classification, 69 (23%) patients had anterior uveitis, 56 (19%) had anterior and intermediate uveitis, 19 (7%) had intermediate uveitis, 15 (5%) had posterior uveitis, and 135 (46%) had panuveitis.

At first visit, 256 (87%) patients were given an EKG and 242 (82%) patients were given a TTE. As part of the diagnostic workup, chest CT scans were done on 282 (96%) patients and considered to be indicative of sarcoidosis in 261 out of 282 (93%) patients.

At baseline, 125 (42.5%) patients had extraocular involvement. This relatively low prevalence of extraocular symptoms was consistent with the high numbers of revelatory uveitis cases at first visit (*n* = 234; 80%). The most common extraocular involvement was symptomatic pulmonary involvement (*n* = 50, 17%), followed by articular and neurological involvements (*n* = 25, 8% each). Cardiac symptoms were present in 8 (3%) patients. EKGs were abnormal in 20 (8%) patients, and TTEs were abnormal in 26 (11%) patients. EKG abnormalities mainly concerned left or right branch block (*n* = 8), first-degree block (*n* = 5), atrial fibrillation (*n* = 3), ventricular hypertrophy (*n* = 3), or complete AVB (*n* = 1). TTE abnormalities concerned hypertrophic cardiomyopathies (*n* = 8), aortic or mitral valvular heart diseases (*n* = 9), or abnormalities consistent with ischemic heart disease (*n* = 2). Overall, most of the EKG or TTE abnormalities were not related to cardiac sarcoidosis.

### 3.2. Evolution

The median duration of follow-up was 60 months (range, 30–104). In total, 239 (81%) patients were followed for more than 2 years in our center. At last visit, all patients were given a complete medical examination; 204 (69%) underwent an EKG and 105 (36%) underwent a TTE. A total of 11 (4%) patients of the cohort underwent cMRI for suspected cardiac sarcoidosis. At time of last visit, 7 (2.4%) patients were determined to have cardiac sarcoidosis, including 3 patients who already had cardiac sarcoidosis at diagnosis of uveitis.

### 3.3. Cardiac Sarcoidosis Involvement

EKG and TTE abnormalities at baseline resulted in the diagnosis of initial cardiac sarcoidosis in 3 (1%) patients. The description of cardiac and extraocular involvement at baseline and during follow-up for the different groups is presented in Table 2. Group 1 was the largest group with 162 (55.1%) patients. Group 2 included 72 (24.5%) patients, and Group 3 concerned 60 (20.4%) patients. In Groups 2 and 3, pulmonary involvement was the most common extraocular involvement (36.1% and 38.3%, respectively). Cardiac involvement was rare: the three patients with cardiac sarcoidosis at first visit had shown previous extraocular involvement at onset of uveitis (Group 3). Abnormal EKG and TTE frequencies were similar between groups, with most abnormalities not appearing to be related to cardiac sarcoidosis after confirmation by cMRI. During follow-up, 48 patients had new organ involvement, including 22 (13.6%) patients in Group 1, 14 (19.4%) in Group 2, and 12 (20%) in Group 3 (*p* = 0.35). Secondary cardiac sarcoidosis was diagnosed in 2 patients in Group 1 and in 2 additional patients in Group 3. At the last visit, 2 (1.2%) patients in Group 1, no patients in Group 2, and 5 (8.3%) patients in Group 3 had cardiac sarcoidosis (*p* = 0.008). cMRI enabled confirmation of cardiac sarcoidosis in the 5 patients by showing typical late gadolinium enhancement (LGE). Only 1 patient diagnosed as cardiac sarcoidosis had a normal cMRI but fulfilled the criteria for probable cardiac sarcoidosis [11] as he presented with treatment-responsive AVB in a context of biopsy proven sarcoidosis, and 1 patient did not undergo a cMRI at diagnosis (context of cardiac arrest).

### 3.4. Description and Outcome of Patients with Cardiac Sarcoidosis

The clinical histories, details of ophthalmological diagnosis, and cardiac and visual outcomes of the seven patients with cardiac sarcoidosis are reported in Table 3. All patients met the cardiac sarcoidosis criteria. The mean age at diagnosis of sarcoidosis was 48.7 (±9.6) years. These cases concerned six females and one male. Two patients were in Group 1, and five were in Group 3. The two patients in Group 1 were younger (33 and 41 years) and presented with anterior and intermediate bilateral uveitis. Both developed multisystemic sarcoidosis with cardiac, cutaneous, and pulmonary involvement. The five patients in Group 3 were 21, 50, 50, 52, and 64 years old, respectively, and presented with unilateral posterior uveitis (2/5), anterior unilateral uveitis (2/5), and posterior bilateral uveitis (1/5). Since five patients were asymptomatic, the diagnosis was based on surveillance EKGs (*n* = 4) or 18-FDG PETs for other indications (*n* = 1). The remaining two patients showed symptoms: one presented with cardiac arrest and the other with lower limb edema and dyspnea. EKGs showed abnormalities in six (85.7%) patients. These were mainly AVB (3/7) and branch blocks (2/7). The TTE was abnormal in two cases, but the cMRI showed late gadolinium enhancement in five patients. Only one patient underwent an endomyocardial biopsy, which pinpointed non-specific lymphocytic infiltration. All patients were treated with a combination of corticosteroids and methotrexate. Three patients received hydroxychloroquine as an additional steroid-sparing agent or for a specific involvement (cutaneous or ocular). No patients received TNF inhibitors or other third-line therapies. One patient required implantation of an ICD for high-grade AVB. On last visit, all seven patients were still alive; five patients showed no cardiac symptoms or inflammations on their cMRIs. One patient had persistent cMRI LGE and one had multifactorial chronic heart failure. Six patients presented a good visual outcome; one suffered from unilateral blindness.

## 4. Discussion

We reported the cardiac outcomes of a large cohort of patients with sarcoid uveitis. After a median follow-up of 5 years, the prevalence of cardiac sarcoidosis was only 2.4%. This low prevalence may be due to our method of patient recruitment, since they are mainly (80%) referred by the Department of Ophthalmology and thus have uveitis as a revelatory symptom. This prevalence is also a consequence of the retrospective design of this study. Indeed, a prospective study with a standardized diagnosis method with cMRI and 18-FDG PET may have detected more paucisymptomatic patients. This frequency is lower than that reported in previous sarcoidosis patient series, where cardiac sarcoidosis was reported in 3–39% of the cases [7,8,9,10,11,12,13]. However, most of these specific studies referred to patients with pulmonary sarcoidosis and symptomatic cardiac involvement.

Cardiac sarcoidosis is a serious disease with a challenging diagnosis and significant morbidity and mortality [17]. In contrast with Han et al. [10], who found 4 of 19 (21%) patients to have severe cardiac sarcoidosis in a case series on sarcoid uveitis, our cohort study reveals relatively favorable cardiac prognosis: five of seven patients with cardiac sarcoidosis experienced no cardiac symptoms or LGE on a cMRI after a first-line treatment. Only one patient required ICD implantation because of cardiac arrest in an initially isolated cardiac sarcoidosis context. An explanation for this difference may reside in the early detection of cardiac sarcoidosis in our cohort and in the application of standardized criteria. Indeed, in our tertiary hospital center, most patients with a diagnosis of uveitis (and a fortiori of sarcoid uveitis) integrate a network of care composed of ophthalmologists and internists. This network may have resulted in early detection of the extraocular extension of sarcoidosis. This appears fundamental, especially in cases of cardiac sarcoidosis where the extent of myocardial damage is directly associated with prognosis [16,17,18]. In particular, most of our patients underwent a structured assessment and standardized follow-up, including a baseline EKG followed by annual EKGs. This strategy enabled detection of cardiac sarcoidosis cases in asymptomatic patients, thus avoiding complications. Despite a lack of sensitivity [12,13,25], EKG is an inexpensive and widely available tool that helps detect cardiac sarcoidosis early on. In 2020, the American Thoracic Society proposed a baseline EKG for all patients with sarcoidosis [26]. A further difference to the study by Han et al. [10] could reside in the population study. Our population was mainly composed of patients mainly of European descent; therefore. our findings may not be generalized to other populations, such as Japanese or African American patients, who could have more severe or frequent disease [15,27,28,29].

TTE was normal in most patients with cardiac sarcoidosis of our study, indicating characteristic abnormalities in only a single patient. This is consistent with the low sensitivity of TTE sensitivity in cardiac sarcoidosis [12,13]. cMRIs were used more often in our study than 18-FDG PET, given that they have good sensitivity and specificity when indicating typical LGE [25]. The prognostic value of the LGE in cMRI is equally important and enables risk stratification in cardiac sarcoidosis, since LGE is linked to death and ventricular tachycardia, while the size of granulomatous infiltrates are linked to poor prognosis [13,18,30,31]. Given such evidence, the latest guidelines have positioned cMRI above TTE or 18-FDG PET for pinpointing suspected cardiac sarcoidosis in patients with extracardiac sarcoidosis [26]. However, 18-FDG PET is valuable to monitor anti-inflammatory treatment in terms of diagnosis in case of contra-indication to cMRI [16].

Risk management and stratification are essential in a multifaceted disease such as sarcoidosis. The heterogeneous presentation, plus the organ involvement and demographic parameters, impede the performance of controlled trials and complicate survival predictions. Scadding proposed one of the first prognostic classifications for sarcoidosis [32]. His score is based on pulmonary involvement and thus scantily is applicable to extrapulmonary sarcoidosis. In recent years, numerous phenotyping studies have attempted to model sarcoidosis prognosis and suggested homogeneous clusters of patients. However, many studies are still based on pulmonary involvement and fail to differentiate extrapulmonary clusters [7,33,34]. In 2018, the GenPhenReSa project proposed five clinical clusters: (1) abdominal organ involvement, (2) ocular-cardiac-cutaneous-central nervous system involvement, (3) musculoskeletal-cutaneous involvement, (4) pulmonary and intrathoracic lymph node involvement, and (5) extrapulmonary involvement [8]. In this multicentric European study involving a cohort of 2163 patients, the patients did not show the same characteristics as those of our cohort, since they were recruited from departments of pneumology and were of European descent only. Nevertheless, our patients with cardiac and ocular disease also had a high proportion of skin and neurological involvement and may correspond to the ocular-cardiac-cutaneous-central nervous system cluster. The association between ocular and neurological clusters is already established; as high as 27% of posterior uveitis cases are associated with CNS involvement [35]. However, the relationship between cardiac and ocular sarcoidosis merits further investigation. One pathophysiological explanation put forward by the authors was that the linking factors of this cluster could be a granulomatous infiltration of the nerves and conduction system of the heart [8].

We tried to stratify the risk of cardiac sarcoidosis by separating patients into three groups as they presented to our services. The simplicity of this classification seems suited to routine clinical practice, as well as risk management, at the bedside or in outpatient clinical centers. The first two groups (clinically isolated ocular sarcoidosis and associated ocular sarcoidosis) showed low prevalence of cardiac sarcoidosis (1.2% and 0%). The third group, containing patients already diagnosed with multisystem sarcoidosis, was at higher risk of cardiac sarcoidosis (8.3%). One hypothesis for the slight prevalence in the first two groups might be that Group 1 had established ocular disease [5] while Group 2 had pre-existing benign extraocular sarcoidosis, discovered on first visit. The multisystem sarcoidosis of Group 3 probably denotes a more aggressive disease. Additionally, there was a trend toward different follow-up durations between Groups 1 and 2 and Group 3 (no statistical significance), although this difference may have favored the development of extraocular involvement in Group 3. Interestingly, patients with cardiac sarcoidosis had systemic sarcoidosis at the time of diagnosing heart involvement, including patients in Group 1 who had progressed to systemic sarcoidosis. The recent official American Thoracic Society Clinical Practice Guideline does not recommend routine cardiac testing after negative baseline screening [26]. Our results suggest the worth of performing additional annual EKGs during follow-up on patients with previous sarcoid uveitis or on those who develop systemic sarcoidosis in order to detect any instances of cardiac sarcoidosis. A prospective study is needed to validate this strategy.

Our study has several limitations. First, it is a monocentric study, conducted in France on a majority of Caucasian patients. Therefore, the results can only be extrapolated to similar populations, as we know that ethnicity plays a role in sarcoidosis heterogeneity [36,37,38]. Second, a retrospective design can be open to errors in data collection, especially since part of the EKG data was not available. The retrospective design may have leaded to underdiagnosis of clinically silent cardiac sarcoidosis as not every patient had cMRI. Third, our cohort was recruited from a tertiary hospital center, resulting in a greater concentration of more severe uveitis cases and possibly causing a selection bias.

## 5. Conclusions

Cardiac sarcoidosis is uncommon and affects 2.4% of sarcoid uveitis patients in this study. Patients with sarcoid uveitis at the revelatory symptom are not likely to develop cardiac sarcoidosis within 5 years of presentation of uveitis. Cardiac sarcoidosis generally occurs in patients with previous symptomatic multisystem involvement or patients progressing from an isolated ocular to a systemic sarcoidosis. Based on these results, we suggest that patients with systemic sarcoidosis and uveitis would benefit from annual clinical and EKG supervision. The systematic repetition of complex cardiac screening tests does not appear to be appropriate for patients with clinically isolated sarcoid uveitis. These results may not be generalized to other populations.

## Figures and Tables

**Table 1 jcm-10-02146-t001:** Characteristics of 294 patients with sarcoid uveitis at diagnosis.

Characteristic	Cohort (*n* = 294)
Age, mean (SD), years	51 (18)
Women, No. (%)	195 (66)
Biopsy-proven, No. (%)	218 (74)
Positive bronchial biopsy, No. (%)	107 (36)
Positive salivary gland biopsy, No. (%)	38 (13)
Positive skin biopsy, No. (%)	32 (11)
Positive biopsy (other sites), No. (%)	41 (14)
Presumed sarcoid uveitis, No. (%)	76 (26)
Uveitis as revelatory symptom, No. (%)	234 (80)
Anterior uveitis, No. (%)	69 (24)
Anterior and intermediate uveitis, No. (%)	56 (19)
Intermediate uveitis, No. (%)	19 (7)
Posterior uveitis, No. (%)	15 (5)
Panuveitis, No. (%)	135 (46)
Bilateral involvement, No. (%)	226 (77)
Extra ocular involvement, No. (%)	125 (43)
Lung involvement, No. (%)	50 (17)
Skin involvement, No. (%)	24 (8)
Neurological involvement, No. (%)	25 (9)
Cardiac involvement, No. (%)	3 (1.0)
High ACE, No. (%)	163 (55)
Compatible chest CT scan, *n*/a (%)	261/282 (93)
Cardiac symptoms, No. (%)	8 (3)
Abnormal EKG, *n*/a (%)	20/256 (8)
Abnormal echocardiogram, *n*/a (%)	26/242 (11)

Data reported as number and percentage, No (%), *n*/a, number/data available. ACE, angiotensin-converting enzyme; EKG, electrocardiogram; IQR, interquartile ranges.

**Table 2 jcm-10-02146-t002:** Cardiac and systemic involvement at baseline and last visit between 3 groups of sarcoid uveitis, defined according to their presentation.

	Group 1 (*n* = 162)	Group 2 (*n* = 72)	Group 3 (*n* = 60)	*p* Values
Number, No. (%)	162 (55.1)	72 (24.5)	60 (20.4)	
Involvement at baseline				
Extra ocular involvement, No. (%)	0 (0)	72 (100)	60 (100)	
Cardiac involvement, No. (%)	0 (0)	0 (0)	3 (5)	
Cardiac symptoms, No. (%)	3 (1.9)	1 (1.4)	3 (5)	
EKG available, No. (%)	143 (88.3)	60 (83.3)	54 (90)	
Abnormal EKG at baseline, *n*/a (%)	7/143 (4.9)	6/60 (10.0)	6/54 (11.1)	
Abnormal echocardiogram at baseline, *n*/a (%)	12/129 (9.3)	9/57 (15.8)	6/55 (10.9)	
Pulmonary involvement, No. (%)	0 (0)	26 (36.1)	23 (38.3)	
Involvement at last visit				
Follow-up period, mean (SD), months	71.2 (55.9)	70.7 (68.4)	96.7 (85.3)	*p* = 0.097
Evolution to systemic sarcoidosis, No. (%)	22 (13.6)			
Cardiac involvement, No. (%)	2 (1.2)	0 (0)	5 (8.3)	*p* = 0.008
Pulmonary involvement, No. (%)	6 (3.7)	33 (45.8)	28 (46.7)	
Skin involvement, No. (%)	5 (3.1)	12 (16.7)	17 (28.3)	
Neurological involvement, No. (%)	2 (1.2)	19 (26.4)	13 (21.7)	
Rheumatologic involvement, No. (%)	4 (2.5)	16 (22.2)	13 (21.7)	
ENT involvement, No. (%)	1 (0.6)	10 (13.9)	2 (3.3)	
Renal involvement, No. (%)	1 (0.6)	3 (4.2)	2 (3.3)	
Hepatic involvement, No. (%)	3 (1.9)	2 (2.8)	6 (10)	

ENT, ear nose throat; EKG, electrocardiogram.

**Table 3 jcm-10-02146-t003:** Details of clinical history of patients with sarcoid uveitis with cardiac involvement.

	Patient 1	Patient 2	Patient 3	Patient 4	Patient 5	Patient 6	Patient 7
Age at diagnosis (y)	41	33	64	50	50	52	21
Sex	Female	Female	Female	Female	Male	Female	Female
Group	1	1	3	3	3	3	3
Uveitis type at diagnosis	Anterior and intermediate bilateral	Anterior and intermediate bilateral	Anterior unilateral	Posterior bilateral	Anterior unilateral	Posterior unilateral	Posterior unilateral
Systemic involvement at first visit	No	No	Cardiac, Cutaneous	Cardiac	Cutaneous, pulmonary, hepatic	Cutaneous	Pulmonary
Ocular sarcoidosis criteria	Biopsy proven (bronchial, skin)	Presumed (ocular signs, BHL, elevated CD4/CD8 ratio)	Biopsy proven (skin)	Biopsy proven (mediastinal lymph node)	Biopsy proven (skin)	Biopsy proven (skin)	Biopsy proven (bronchial)
CS criteria	Histological diagnosis of extracardiac sarcoidosis + high grade atrioventricular block resolutive under treatment	Ophthalmic sarcoidosis + LGE on CMRI + Basal thinning of the ventricular septum + elevated ACE and lysosyme + BHL + lymphocytic alveolitis	Histological diagnosis of extracardiac sarcoidosis + LGE on CMRI	Histological diagnosis of extracardiac sarcoidosis + unexplained high-grade atrioventricular block	Histological diagnosis of extracardiac sarcoidosis + LGE on CMRI	Histological diagnosis of extracardiac sarcoidosis + LGE on CMRI	Histological diagnosis of extracardiac sarcoidosis + LGE on CMRI
Period between uveitis and cardiac involvement (months)	48	144	1	−36 (CS as presenting symptom)	−18	72	57
Systemic involvement at last visit	Cutaneous sarcoid, Pulmonary, cardiac	Cardiac, pulmonary	Cardiac	Cardiac, hypercalcemia	Cardiac, renal, pulmonary, cutaneous, hepatic	Cardiac, neurologic, cutaneous	Cardiac, Neurologic, pulmonary
Symptoms of CS	No, Routine EKG	Dyspnea, edema lower limbs	No, Routine EKG	Cardiac arrest	No, Routine EKG	No, Routine EKG	No
EKG at diagnosis of CS	Second degree AVB	Sinusal, repolarization abnormalities	First degree AVB	Complete AVB, after resuscitation	Bifascicular block	Left branch block	Normal
Echocardiography	Normal	Good LVEF, basal ventricular septum dyskinesia, and thinning	Normal	NA	Normal	Normal	Normal
CMRI	Normal	Septal LGE	Septal LGE	NA	Septal LGE	Septal LGE, septal dyskinesia	Septal LGE, septal dyskinesia
EMB	NA	NA	NA	non-specific lymphocytic infiltration	NA	NA	NA
Treatment	Corticosteroids, MTX	Corticosteroids, MTX	Corticosteroids, MTX	Corticosteroids, MTX, hydroxychloroquine	Corticosteroids, MTX, hydroxychloroquine	Corticosteroids, MTX, hydroxychloroquine	Corticosteroids, MTX
ICD	No	No	No	Yes	No	No	No
Follow up (months)	74	192	30	240	132	96	204
Outcome at last visit	normalization of EKG under treatment	normalization of cardiac abnormalities, development of pulmonary hypertension	no cardiac symptoms	chronic cardiac insufficiency	asymptomatic	stability, persistent cardiac inflammation	no cardiac symptoms or inflammation

AVB, atrioventricular block; BHL, Bilateral hilar lymphadenopathy; CMRI, Cardiac magnetic resonance imaging; CS, cardiac sarcoidosis; EKG, electrocardiogram; EMB, endomyocardial biopsy; ICD, implantable cardioverter defibrillator; LGE, late gadolinium enhancement; LP, light perception; LVEF, left ventricular ejection fraction; MTX, Methotrexate.

## Data Availability

Data are available upon request to the corresponding author.

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
