# Peer review of "Cardiac Sarcoidosis Is Uncommon in Patients with Isolated Sarcoid Uveitis: Outcome of 294 Cases"

_jcm, 2021, doi:10.3390/jcm10102146_

Round 1

Reviewer 1 Report

The initial review concerns have been addressed adequately.

Author Response

The reviewer did not raise any further concern.

Reviewer 2 Report

  1. The comments that I have previously raised have largely been addressed.

I make some further comments.

  1. Has enough data been supplied to confirm that all the patient is under consideration fulfil the criteria for presumed or definite sarcoidosis related uveitis?  I think yes.

Table one does set out the number of patients in each group

In the biopsy-proven group however, the biopsies mentioned do not add up to 218  - only 117.

Presumed sarcoid uveitis is stated – 26%

  1. I am happy with the definitions of groups 1 & 2

However I do not fully understand group 3 – “Group 3: systemic sarcoidosis (patients with extraocular involvement before the onset of uveitis”.  At what point did these patients develop uveitis -it does not seem to be clear. Groups 1 in 2 all had uveitis at the baseline but not group 3

  1. Table 2 is initially confusing as it has 3 parts which I think need to be more clearly separated. It is however a way of showing the progression of disease.

May be it needs to be stated that a percentage of group one evolves to be not just clinically isolated sarcoid uveitis anymore. 

Perhaps separating table 2 into 3 separate tables with some brief explanation of each would be helpful

  1. Conclusion statement:
    I do not think the reworking of this statement is correct.

Patients with sarcoid uveitis as the revelatory symptom is group 1

In this group it appears that 2 patients were shown to have cardiac involvement within the first year of follow up. What would seem to be important is recognising when the patients who were clinically isolated sarcoid uveitis,  have become multisystem patients and then need to be screened . Stating the incidence as 1.2% for group 1 is strictly not true, as I understand it, as they have evolved to multisystem disease and you could argue they transition to group  2.

(Have I got this correct?)

Your statement is correct IF they remain clinically isolated. So maybe reword  Lines 289 – 290.

Certainly agree the multi system disease group needs closer monitoring.

Your deleted statement “that cardiac sarcoidosis generally occurs in patients with previous symptomatic multisystem involvement”  would seem to be true as this is groups 2 & 3. So why delete it?

  1. As far as the title is concerned -I think the alteration makes it more meaningful

Author Response

POINT 1: Has enough data been supplied to confirm that all the patient is under consideration fulfil the criteria for presumed or definite sarcoidosis related uveitis?  I think yes.

Table one does set out the number of patients in each group

In the biopsy-proven group however, the biopsies mentioned do not add up to 218  - only 117.

Presumed sarcoid uveitis is stated – 26%

 Response 1: Indeed, we added in the Table 1 the missing patients (41 patients, 14%) with biopsy proven sarcoidosis from various biopsy site (peripheral lymph node, kidney, liver, conjonctiva). In addition, we simplified the percentages in the table.

POINT 2 : I am happy with the definitions of groups 1 & 2

However I do not fully understand group 3 – “Group 3: systemic sarcoidosis (patients with extraocular involvement before the onset of uveitis”.  At what point did these patients develop uveitis -it does not seem to be clear. Groups 1 in 2 all had uveitis at the baseline but not group 3.

Response 2 : Group 3 consists of patients with diagnosed sarcoidosis (pulmonary, neurologic…) initially without ocular involvement, who will develop uveitis. We changed the definition: “Group 3: systemic sarcoidosis (patients diagnosed with systemic sarcoidosis before the onset of uveitis).” Line 106

POINT3: Table 2 is initially confusing as it has 3 parts which I think need to be more clearly separated. It is however a way of showing the progression of disease.

 Maybe it needs to be stated that a percentage of group one evolves to be not just clinically isolated sarcoid uveitis anymore. 

 Perhaps separating table 2 into 3 separate tables with some brief explanation of each would be helpful

Response 3 :  We agree with the reviewer that the table 2 is confusing. The middle part is unnecessary, so we modified the table by specifying only the beginning and the end of the follow-up and integrated the information of systemic evolution of group 1 in the final part of the table. Table 2 is thus clearer.

POINT 4: Conclusion statement:
I do not think the reworking of this statement is correct.

Patients with sarcoid uveitis as the revelatory symptom is group 1

In this group it appears that 2 patients were shown to have cardiac involvement within the first year of follow up. What would seem to be important is recognising when the patients who were clinically isolated sarcoid uveitis,  have become multisystem patients and then need to be screened . Stating the incidence as 1.2% for group 1 is strictly not true, as I understand it, as they have evolved to multisystem disease and you could argue they transition to group  2. (Have I got this correct?) Your statement is correct IF they remain clinically isolated. So maybe reword  Lines 289 – 290.

Certainly agree the multi system disease group needs closer monitoring.

Your deleted statement “that cardiac sarcoidosis generally occurs in patients with previous symptomatic multisystem involvement”  would seem to be true as this is groups 2 & 3. So why delete it?

RESPONSE 4: The reviewer is correct, the patient with cardiac sarcoidosis in Group 1 had systemic symptoms at the time of their cardiac involvement. We reworked the statement as follows: “Cardiac sarcoidosis is uncommon and affects 2.4% of sarcoid uveitis patients in this study. Patients with sarcoid uveitis at the revelatory symptom are not likely to develop cardiac sarcoidosis within 5 years of presentation of uveitis. Cardiac sarcoido-sis generally occurs in patients with previous symptomatic multisystem involvement or patients progressing from an isolated ocular to a systemic sarcoidosis. Cardiac sar-coidosis generally occurs in patients with previous symptomatic, multisystem in-volvements. Based on our this results, we would suggest that patients with systemic sarcoidosis and uveitis would benefit from annual clinical and EKG-based supervision. The systematic repetition of complex cardiac screening tests does not appear to be ap-propriate for patients with clinically isolated sarcoid uveitis. These results may not be generalized to other populations.”

POINT 5: As far as the title is concerned -I think the alteration makes it more meaningful

Response 5 : We agree with the reviewer opinion.

This manuscript is a resubmission of an earlier submission. The following is a list of the peer review reports and author responses from that submission.

Round 1

Reviewer 1 Report

Overall I think this is a really useful paper that deserves a wide readership audience including physicians and ophthalmologists.
Although the study is retrospective and has limitations as pointed out, it justifies publication as it help stratify patients who present with ocular sarcoidosis as well as indicating prevalence.

I make the following comments that should be considered:

Line 89: Starts with cardiac symptoms but should go on to separately address definitions of cardiac investigations

Line 92: The word “retained” is probably incorrect and should be changed to “diagnosed”

Line 136: I cannot find the percentage of patients in the study group who had a cMRI, and TTE. You quote for EKG. It is alluded to later in a paper that some patients did not but that this should be defined more clearly.

Line 155: It is not clear on what basis definite cardiac sarcoidosis was made. I am presuming they had a cMRI or PET??

Line 174: It would be good to know on what basis the one patient diagnosed with cardiac sarcoidosis who had a normal cMRI, was determined to have cardiac sarcoidosis

Line 212: It would be good at this point to mention the value of PET scanning which may have also picked up some other patients where cardiac sarcoidosis was suspected.

Line 248: There is a very good review I have quoted below. It is not referenced although the author has been referenced in another way. This reference is possibly better than the one you have quoted (16).

Cardiovasc Diagn Ther 2016;6(1):50-63  Edward Hulten etal . Cardiac sarcoidosis—state of the art review.

Line 268: Reference (35) has been missed quoted and should read “… as high as 27%”

Line 301: It would be better stated “….and affects 2.4% of the sarcoid uveitis patients in this study”

Line 305: I would like to see it more clearly stated that an initial investigation of all patients should include an EKG and this should be used as you say for longer term supervision.

I quite liked the discussion in the article I quoted above by Hulten:

“18F-FDG PET offers a reasonable alternative to CMR, although it does expose patients to ionizing radiation and does require careful patient preparation. On the other hand, for patients with glomerular filtration rate below 30 mL/min, ferromagnetic devices such as defibrillators or pacemakers (with the exception of a few approved devices), or other contra-indication to CMR, imaging with 18F-FDG PET may be useful. 67Ga scanning was traditionally offered and is still included in the 2006 modified JMH criteria for cardiac sarcoidosis, but this test is not commonly performed in the United States and is not considered first line due to its inferior accuracy when compared to perfusion and 18F-FDG PET (48).

Perhaps referencing this could be useful and also the flow diagram summarises a pathway of care which you have alluded to. This might help you clarify your recommendations .

Reviewer 2 Report

There are several issues with  the paper that dampen the enthusiasm of this reviewer.  First and foremost is that the title is misleading.  It should at minimum clarify that cardiac is uncommon in people for whom Uveitis was the presenting and in most cases the ONLY presentation of sarcoidosis.  Which leads to the second major point that it is hard to be convinced that these patients are truly sarcoidosis with such low prevalence of lung manifestations.  The authors need to clarify several things like: 1) for those biopsy proved, from what tissue were that biopsy taken?  2) for those patients with "extra-ocular" involvement what other organs were involved (since so few had lung involvement); again, this speaks to how convincingly the results are applicable to systemic sarcoidosis which is  the disease of greatest concern.  The authors also need to clarify their study  design and the way it is described.  They say it is a retrospective study, but talk repeatedly about  follow-up.  After multiple readings it seems that they  "followed patients in time" through the medical records, but again, this is  not clear.  I feel that the overall conclusions of the study are not comparable  to previous  reports and the  authors need to clarify the main finding which is: Patients with uveitis indicative of sarcoidosis at the revelatory symptom are not likely to develop either systemic, cardiac or pulmonary sarcoidosis within 5 years of presentation of uveitis."  This is a different conclusion than what the title indicates.  The results would be quite different if examining a cohort of persons with other primary organ  involvement who also had uveitis.